# Neuropathic-like Pain Symptoms and Their Association with Muscle Strength in Patients with Chronic Musculoskeletal Pain

**DOI:** 10.3390/jcm11185471

**Published:** 2022-09-17

**Authors:** Hee Jung Kim, Min Gi Ban, Kyung Bong Yoon, Woohyuk Jeon, Shin Hyung Kim

**Affiliations:** 1Department of Anesthesiology and Pain Medicine, Anesthesia and Pain Research Institute, Yonsei University College of Medicine, Seoul 03722, Korea; 2Department of Anesthesiology and Pain Medicine, Yongin Severance Hospital, Yonsei University College of Medicine, Yongin 16995, Korea

**Keywords:** chronic pain, hyperalgesia, muscle strength, musculoskeletal pain, sarcopenia

## Abstract

The relationship between sarcopenia and pain remains unclear; thus, this study evaluated whether muscle strength is independently associated with neuropathic-like pain symptoms in patients with chronic musculoskeletal pain. A cut-off score of painDETECT ≥13 was used to indicate a possible neuropathic component. Handgrip strength was measured, and muscle mass was estimated. A total of 2599 patients, including 439 patients who reported neuropathic-like pain symptoms (16.9%), were included for analysis. Handgrip strength was significantly lower in patients experiencing neuropathic-like pain symptoms (23.23 ± 10.57 vs. 24.82 ± 10.43 kg, *p* < 0.001), and this result was chiefly found in female patients. However, there was no difference in estimated muscle mass. Shorter duration of pain, opioid usage, pain in lower limbs, sleep disturbance, and lower handgrip strength were significantly associated with neuropathic-like pain symptoms. In patients with handgrip strength below the reference values by sex, experiencing radiating pain and at least moderate sensory symptoms by light touch and thermal stimulation were more frequently reported. In conclusion, lower handgrip strength appeared to be an independent factor associated with symptoms suggestive of neuropathic pain in this population. Interventional studies are required to determine whether improvement in muscle strength can reduce the neuropathic pain component in chronic musculoskeletal pain.

## 1. Introduction

Sarcopenia is a geriatric syndrome characterized by a decrease in muscle mass, muscle strength, and physical performance. According to a recent systematic review and meta-analysis, the prevalence of sarcopenia ranged from 10 to 27%, although there were differences according to classification criteria or cut-off values. Untreated sarcopenia causes functional degradation and disability, impacts quality of life, and requires increased costs of care [1,2,3,4]. Handgrip strength (HGS) is the most commonly used measure for global muscle strength, and obtaining HGS is the first step in diagnosing sarcopenia [1,2,3]. Low HGS is a clinically relevant predictor of poor patient outcomes such as longer hospitalization, impaired functional status, mental health problems, poor quality of life, and mortality [5,6,7,8].

Among patients with chronic pain, a fifth of patients appear to experience neuropathic pain symptoms which are potentially caused by a lesion or disease affecting the somatosensory nervous system [9,10]. The burden of neuropathic pain components in chronic pain patients seems to be related to the complexity of pain symptoms and poor treatment outcomes. Patients suffering from neuropathic pain showed impaired quality of life due to socioeconomic problems, as well as morbidity from the pain itself and the inciting disease [11]. Preoperative neuropathic-like pain symptoms were associated with poor outcomes in terms of pain relief after total knee replacement surgery [12]. Therefore, early detection of a neuropathic pain component in chronic pain should be important in clinical practice. The self-reported painDETECT questionnaire is one of the validated screening tools for evaluating the presence of neuropathic pain components in chronic pain disorders [13,14]. The painDETECT questionnaire was developed to identify neuropathic component in low back pain, and it has been performed to estimate the prevalence of neuropathic pain component in a wide range of chronic pain conditions [15].

The causal relationship between sarcopenia and pain remains unclear, but patients with pain appeared at a higher risk of sarcopenia than those without pain [16]. The fear of pain following movement or physical activity in patients with musculoskeletal pain can be a predisposing factor relevant to the development and aggravation of loss of muscle strength and mass. A recent longitudinal study identified sarcopenia as one of the risk factors for the development of new neuropathic pain symptoms in healthy middle-aged and elderly subjects [17]. Although there are differences between the selected studies, the prevalence of chronic musculoskeletal pain is reported to be around 15–45%, which lowers the quality of life of patients and causes high socioeconomic costs [18]. However, there have been few large studies to examine the impact of sarcopenia-related factors on the presence of neuropathic pain components and its pain characteristics in the subgroup of patients with chronic musculoskeletal pain. We hypothesized that reduced muscle strength or muscle mass might be associated with neuropathic-like pain symptoms in patients with chronic musculoskeletal pain.

Therefore, the goal of this study was to explore the association between HGS and neuropathic-like pain symptoms based on painDETECT results in patients with chronic musculoskeletal pain. In addition, the factors associated with neuropathic-like pain symptoms in this population were investigated, and differences in the characteristics of pain symptoms according to HGS, based on the results of each painDETECT item, were examined.

## 2. Materials and Methods

### 2.1. Study Population

Approval from the Institutional Review Board was obtained and in keeping with the policies for a retrospective review, informed consent was not required (No. 4-2022-0168). A total of 3712 adult patients aged 20 to 98 who underwent treatment for their pain at the pain clinic of a tertiary university hospital in 2017 were initially included in this study. Musculoskeletal pain was defined as pain in the following five anatomical areas: neck (cervical spine); shoulder and upper limbs (elbow, wrist, and hand); back (thoracic spine) and chest wall; lower back (lumbar spine); and hip and lower limbs (knee, ankle, and foot). Chronicity was established by the persistence of pain beyond 3 months. Patients with incomplete medical records or who could not measure handgrip strength due to hand osteoarthritis or a neurological disorder were excluded. Additionally, patients with acute pain (<3 months) or who did not have musculoskeletal pain were excluded.

### 2.2. PainDETECT Score Measurements

Neuropathic-like symptoms were assessed by means of the patient-reported painDETECT questionnaire at the patient’s initial visit to our clinic. Patients freely described their pain symptoms and filled out the validated Korean version of the painDETECT questionnaire [19]. The painDETECT questionnaire is composed of a main item along with two additional items. In the main component, seven 0 to 5 Likert-scaled questions regarding pain quality with descriptions of different sensations were asked: burning, tingling or prickling sensations, tactile allodynia, thermal allodynia, electric shock-like sensations, numbness, and pressure-evoked pain sensation (0 = never; 1 = hardly noticed; 2 = slightly; 3 = moderately; 4 = strongly; 5 = very strongly). For analysis, the responses were dichotomized into moderate to severe sensory symptoms (score of ≥3 on each question) relative to the combined other responses (score of <3 on each question). In addition, pain course patterns were investigated as follows: persistent pain with slight fluctuations (0 points); persistent pain with pain attacks (−1 point); pain attacks without pain between them (1 point); and pain attacks with pain between them (1 point). Lastly, radiation of pain to other regions of the body (2 points), which is a neuropathic descriptor, was evaluated. The painDETECT questionnaire yields a minimum and maximum total score of −1 and 38, respectively, with higher scores being suggestive of neuropathic-like pain. For the purpose of this study, a cutoff at 13 points was used to discriminate a possible or highly likely neuropathic pain phenotype (painDETECT score ≥ 13) from an unlikely neuropathic pain phenotype (painDETECT score < 13) [17,20]. Thus, the presence of neuropathic pain components, reporting neuropathic-like pain symptoms, was defined as cases with a score of ≥ 13 on the painDETECT questionnaire.

### 2.3. Handgrip Strength Measurement and Muscle Mass Estimation

HGS was measured three times each on the left and right side using a Smedley-type handheld dynamometer (EH101; CAMRY, Guangdong, China) at the initial visit. HGS was measured and recorded by an independent tester. The patients were requested to sit in a comfortable position with their elbows extended and to squeeze the dynamometer with maximum strength [21]. The highest reading among the three trials was used for the analysis [21]. We also analyzed muscle mass using the appendicular skeletal muscle mass (ASM) and the skeletal muscle mass index (SMI). ASM was estimated using a previously validated anthropometric equation for the Asian population: ASM (kg) = 0.193 × body weight (kg) + 0.107 × height (cm) − 4.157 × sex (men = 1 or women = 2) − 0.037 × age (years) − 2.631 [22,23]. The SMI was calculated as ASM divided by the square of height in meters.

### 2.4. Patient Demographics and Clinical Data Measurements

Patient characteristics including age, sex, body mass index (BMI), and pain-related variables were collected by electronic medical record chart review. Variables included diagnosed comorbidities that led to continued medical interventions with regular hospital visits, such as hypertension, diabetes mellitus, cardiovascular disease, mental health problems, and osteoporosis/osteopenia. Mental health problems were defined as cases experiencing psychiatric treatment or medication for depression, anxiety, or stress in the past year. The following factors were identified as pain related: duration of pain, average pain score on the numeric rating scale (NRS, 0 to 10) in the last 4 weeks, current opioid usage for at least 1 month, primary pain locations according to anatomical area groupings, the presence of multiple pain sites, and sleep disturbance. The presence of multiple pain sites was defined as pain in at least two different anatomic body sites. Sleep disturbance was defined as any difficulty in the initiation of sleep and/or awakening midnight and use of sleep pills over the course of at least 3 months. 

### 2.5. Statistical Analysis

Data are expressed as mean ± standard deviation (SD) for continuous variables and number (percentages) for categorical variables. For ordinal data and discontinuous variables, we expressed the data as median and interquartile range (IQR). The normality of distribution was examined by the Shapiro–Wilk test to ensure proper statistical treatment. Demographics and clinical variables were analyzed by independent *t*-test, chi-squared test, or Fisher’s exact test as appropriate. The Mann–Whitney U test was utilized for continuous variables with non-normal distributions. Significant univariate variables with a *p*-value threshold of 0.2 were included in multivariate logistic regression analyses to identify the factors associated with neuropathic-like symptoms based on a painDETECT score of ≥13 in patients with chronic musculoskeletal pain, and the adjusted odds ratios (aORs) and 95% confidence intervals (CIs) were calculated. For a sub-analysis, the study population was divided into two groups: a normal HGS group (≥28 kg for men and ≥18 kg for women) and a low HGS group (<28 kg for men and <18 kg for women), according to the Asian Working Group for Sarcopenia (AWGS) 2019 guideline [1]. The results of the painDETECT questionnaire items were compared between the two groups. All statistical analyses were conducted using the SPSS, version 26.0 (IBM Corp, Armonk, NY, USA). A *p*-value < 0.05 was considered statistically significant. 

## 3. Results

Overall, a total of 2599 patients aged 20 to 98 years, consisting of 439 patients with possible neuropathic pain components (painDETECT score ≥ 13) and 2160 patients unlikely to have neuropathic pain components (painDETECT score < 13), were finally analyzed in this study (Figure 1). 

Patient demographics, pain-related clinical data, and sarcopenia-related data were compared between the two groups (Table 1). Age, sex, and BMI did not differ significantly between the two groups, also painDETECT scores were similar regardless of age. The prevalence of hypertension, diabetes mellitus, cardiovascular disease, and osteopenia/osteoporosis was similar in both groups. However, the proportion of mental health problems was significantly higher in the group with neuropathic-like pain symptoms. The median duration of pain was significantly shorter in patients with neuropathic-like pain symptoms than in those without neuropathic-like pain symptoms (8 vs. 12 months, *p* < 0.001). The mean pain score was significantly higher in patients with neuropathic-like pain symptoms than in those without neuropathic-like pain symptoms (NRS: 6.71 ± 2.09 vs. 6.02 ± 2.16, *p* < 0.001). In addition, the proportions of patients taking opioids and experiencing sleep disturbances were significantly higher in patients with neuropathic-like pain symptoms. The lower back area was the most common pain location regardless of the presence of neuropathic-like pain symptoms, and there was a significant difference in pain locations between the two groups. Patients with pain on back or chest wall and lower limbs reported higher painDETECT scores. Patients with pain on upper limbs or low back area showed lower HGS than those with other pain sites. The prevalence of multiple pain sites was comparable between the two groups. The sarcopenia-related data, ASM and SMI, were comparable between the two groups. However, HGS was significantly lower in patients with neuropathic-like pain symptoms (23.23 ± 10.57 vs. 24.82 ± 10.43 kg, *p* < 0.001).

The sex-specific comparison showed that HGS was significantly lower in female patients with neuropathic pain components, but not in male patients. A significant difference was not observed in estimated muscle mass parameters, both ASM and SMI, between the two groups for both sexes (Table 2).

Multivariate logistic regression analysis showed that shorter duration of pain (aOR 0.536; 95% CI 0.390–0.738; *p* < 0.001), opioid usage (aOR 2.961; 95% CI 1.860–4.714; *p* < 0.001), pain in lower limbs (aOR 2.066; 95% CI 1.102–3.872; *p* = 0.024), sleep disturbance (aOR 2.052; 95% CI 1.487–2.831; *p* < 0.001), and lower HGS (aOR 0.976; 95% CI 0.960–0.993, *p* = 0.005) were independent factors associated with neuropathic-like pain symptoms (Table 3). 

When comparing the results of each of the painDETECT items between the normal HGS (n = 1621) and the low HGS group (n = 978), the proportion of patients experiencing moderate to severe sensory symptoms (score of 3 ≥ 5 on each question) resulting from either light touch or thermal stimulation was significantly higher in the low HGS group. In addition, there was less persistent pain with slight fluctuation, and a radiating pain pattern was more predominantly reported, in patients with low HGS. The total painDETECT score was significantly higher in patients with low HGS. Other characteristics of pain symptoms on painDETECT were similar between the two groups (Table 4).

## 4. Discussion

Several demographic (older age, female), psychosocial (depression, anxiety, disturbed sleep, alcohol, smoking, overweight), clinical (preexisting pain, diabetics, hyperlipidemia), and genetic factors have been reported to be associated with neuropathic pain [24]. Additionally, multisite pain such as fibromyalgia and emotional distress were related to neuropathic symptoms in patients with chronic musculoskeletal pain [25]. However, these previous reports did not include any sarcopenia-related clinical parameters. Although a cause-and-effect relationship could not be established in this study, our results demonstrated that lower HGS, as an indicator of potentially reduced global muscle strength, was significantly associated with neuropathic-like pain symptoms in patients with chronic musculoskeletal pain. 

The causal relationship between pain and sarcopenia has not been fully elucidated; however, a positive correlation between them was found in a recent meta-analysis based on results of 10 observational studies [16]. Currently, loss of muscle strength and function, rather than muscle mass, has been emphasized in the diagnosis of sarcopenia [2]. Indeed, previous studies showed that a significant decline in muscle strength was found for fibromyalgia patients compared with healthy controls, whereas the association was not observed for loss of muscle mass [26,27]. As shown in our results, chronic musculoskeletal pain patients with neuropathic pain components reported a higher pain intensity. This might lead to higher opioid use in patients with neuropathic pain components. Our results also showed that pain in the lower limbs, which can affect gait disturbance, was associated with neuropathic-like symptoms, similar to a previous report [28]. Kinesiophobia is frequent in patients with chronic musculoskeletal pain [29]. There are also several studies showing that the incidence of chronic pain is lower in physically active people [18]. Therefore, patients with higher neuropathic pain components are more likely to experience lower physical performance. This situation can be a predisposing factor relevant to the development and aggravation of loss of muscle strength. Additionally, chronic low-grade inflammation in severely impaired chronic pain patients can contribute to impaired muscle strength [30,31]. In the concept of dynapenia, several neurophysiological mechanisms, such as a decreased excitatory drive from supraspinal centers, a decreased α-motoneuron excitability, and a reduced motor unit recruitment and rate coding, have been proposed to explain the loss of muscle strength with aging [32]. Persistent and severe pain in the musculoskeletal system showed the alteration of the neuromuscular system similar to this aging-related muscle weakness. In human subjects, progressively increased muscle pain intensity resulted in a gradual decrease in motor unit firing rates [33]. Additionally, high fat content in the paraspinal muscles, but not reduced muscle mass, was closely related to severe pain and poor functional status and structural changes of the lumbar spine region in patients with chronic back pain [34]. Taken together, these findings indicate that chronic musculoskeletal pain patients experiencing more neuropathic-like pain symptoms are potentially more vulnerable to loss of muscle strength. Conversely, sarcopenia has been identified as one of the risk factors for the development of new neuropathic pain symptoms in the healthy population [17]; thus, locally or systemically reduced muscle strength in patients with chronic musculoskeletal pain can contribute to newly developed or exacerbated neuropathic-like pain symptoms. This proposed explanation may be supported by previously reported clinical evidence, in which exercise programs including resistance training reduced the risk of developing neuropathic pain and reduced pain intensity in diabetic neuropathy, chemotherapy-induced peripheral neuropathy and fibromyalgia [35,36,37].

In the present study, female patients with neuropathic pain components interestingly exhibited lower HGS than those without with neuropathic pain components, but this was not observed in male patients. The overall prevalence of sarcopenia and chronic pain was higher in women compared to men [16,24,38]. Similar to our result, female patients with knee osteoarthritis are more likely to experience neuropathic-like pain symptoms than male patients [39]. Sex hormone affects maintaining skeletal muscle homeostasis [40]. Muscle strength progressively decreases after menopause, and higher HGS was associated with a lower risk of poor health-related quality of life in postmenopausal women [41]. There seems to be a lack of information on sex differences between muscle strength and neuropathic pain. Stronger consideration should be given to study this sex difference and to integrate the findings into clinical approaches to pain management.

In addition, our results showed that shorter pain duration was associated with neuropathic-like pain symptoms. In a previous study in which similar scores were observed among musculoskeletal pain patients with several months to years of pain duration, the painDETECT score was not associated with the time elapsed since pain onset [42]. However, this may not be inconsistent because classifications of pain duration and the types of neuropathic pain screening tools used according to the study design can affect the result [28]. Much of the study population was suffering from comorbid centrally medicated symptoms such as some mental health problems and sleep disturbance. Among them, sleep disturbance was significantly associated with neuropathic-like symptoms in this study. Previous reports have consistently demonstrated that patients with chronic neuropathic pain suffered from anxiety, depression, and poor sleep quality, symptoms that are shared with central mechanisms of pain [10,13,43]. A causal bidirectional relationship between sleep disturbance and joint pain with neuropathic features, but not with other types of joint pain, was observed in patients who underwent total joint replacement [44]. Neuropathic pain management remains challenging, and part of the challenge may result from the heterogeneity of symptoms and the lack of pain profiling [45]. In this study, patients with low HGS below the reference values by sex more frequently reported stimulus-evoked positive sensory symptoms suggestive of allodynia and thermal hyperalgesia than spontaneous sensory symptoms such as burning and tingling/prickling. This result suggests that sensory symptoms in patients with low HGS appear in specific symptom clusters that may underlie a higher level of neuroplastic changes including cognitive-emotional sensitization [46,47]. 

There are several limitations in this study. There may be information bias due the retrospective nature of the current study. Additionally, this study was performed in a single-center institution with a sample involving a homogenous racial study population. This cross-sectional study could not determine a causal relationship between muscle strength and neuropathic pain; thus, a longitudinal study with a large population should be conducted. In this study, socioeconomic factors, which can affect pain and sarcopenic status, were not included. This study used a real-world clinical practice model; thus, most patients were already taking various types of analgesics or received some interventional pain procedures during the initial visit to our clinic. This potential confounder, especially exposure to gabapentinoids and antidepressants, could affect the painDETECT score. In this study, the detailed diagnoses of the study population based on pain etiologies could not analyzed due to some technical difficulties. Although ASM and SMI were estimated using the previously validated equation, this equation has not been validated in patients with chronic pain. Additionally, bioelectrical impedance analysis or dual-energy X-ray absorptiometry were not used in this study. Thus, low HGS alone did not fully satisfy the current diagnosis criteria for sarcopenia in the present results. Finally, we used a single self-reported questionnaire in this study; therefore, evidence of the association of muscle strength with neuropathic pain using neurophysiological tests may be warranted.

## 5. Conclusions

This study demonstrated that lower HGS was an independent factor associated with neuropathic-like pain symptoms in patients with chronic musculoskeletal pain. This was predominantly observed in female patients, and patients with low HGS below the reference values by sex reported more sensory symptoms suggestive of allodynia and thermal hypersensitivity. Therefore, by performing HGS measurements for patients with chronic musculoskeletal pain, it is possible to expect the development of neuropathic-like pain symptoms and provide appropriate management to increase the probability of a positive outcome. Furthermore, interventional studies are required to determine whether improvement in muscle strength can reduce neuropathic pain components in chronic musculoskeletal pain.

## Figures and Tables

**Figure 1 jcm-11-05471-f001:**
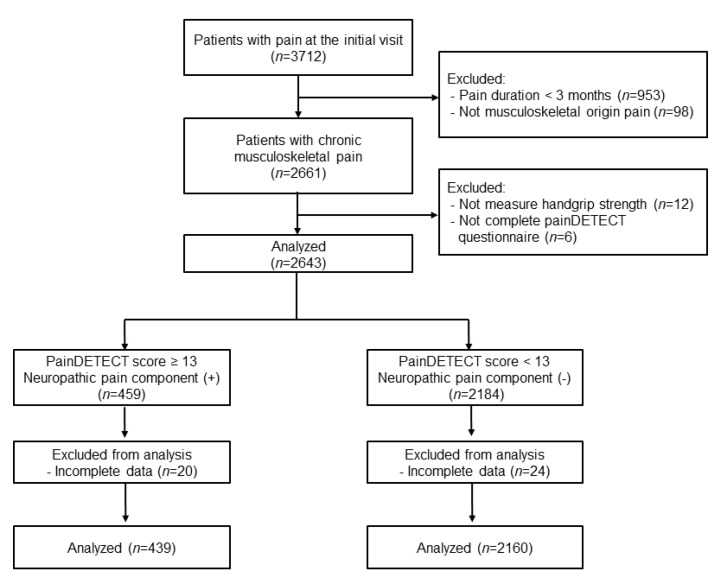
Flow chart.

**Table 1 jcm-11-05471-t001:** Characteristics, pain-related data, and sarcopenia-related data between patients who reported a painDETECT score ≥ 13 (possible neuropathic-like pain symptoms) and those with a painDETECT score < 13 in patients with chronic musculoskeletal pain.

Variables	PainDETECT ≥ 13 (*n* = 439)	PainDETECT < 13 (*n* = 2160)	*p*-Value
Patient characteristics			
Age, years	59.46 ± 16.03 (20–98)	60.02 ± 15.50 (20–89)	0.487
<45 years	82 (18.7)	352 (16.3)	0.393
45–64 years	171 (39.0)	874 (40.5)	
≥65 years	186 (42.3)	934 (43.2)	
Sex, M/F	165 (37.6)/274 (62.4)	894 (41.4)/1266 (58.6)	0.139
BMI, kg/m^2^	24.32 (21.91–26.01)	24.03 (22.06–26.03)	0.971
<25 kg/m^2^	276 (62.9)	1281 (59.3)	0.347
≥25 kg/m^2^	163 (37.1)	879 (40.7)	
Medical comorbidities, n			
Hypertension	131 (29.8)	664 (30.7)	0.755
Diabetes mellitus	81 (18.5)	410 (19.0)	0.830
Cardiovascular disease	19 (4.3)	128 (5.9)	0.193
Mental health problems	136 (31.0)	545 (25.2)	0.010
Osteopenia/osteoporosis	147 (33.5)	723 (33.5)	0.943
Pain-related data			
Pain duration, months	8.00 (3.00–36.00)	12.00 (6.00–48.00)	<0.001
<12 months	238 (54.2)	813 (37.6)	<0.001
≥12 months	201 (45.8)	1347 (62.4)	
Pain score, NRS 0–10	6.71 ± 2.09	6.02 ± 2.16	<0.001
NRS < 7	185 (42.1)	1176 (54.4)	<0.001
NRS ≥ 7	254 (57.9)	984 (45.6)	
Opioid usage, n	80 (18.2)	137 (6.3)	<0.001
Primary pain location, n			<0.001
Neck (cervical spine)	80 (18.2)	373 (17.3)	
Upper limbs (shoulder, elbow, wrist, hand)	46 (10.5)	172 (8.0)	
Back (thoracic spine), chest wall	58 (13.2)	158 (7.3)	
Lower back (lumbar spine)	197 (44.9)	1298 (60.1)	
Lower limbs (hip, knee, ankle, foot)	58 (13.2)	159 (7.4)	
Multiple pain sites, n	118 (26.9)	599 (27.7)	0.716
Sleep disturbance, n	254 (57.9)	816 (37.8)	<0.001
Sarcopenia-related data			
ASM, kg	17.74 ± 4.79	18.30 ± 4.67	0.119
SMI, kg/m^2^	6.71 ± 1.17	6.87 ± 1.15	0.082
HGS, kg	23.23 ± 10.57	24.82 ± 10.43	0.004

Values are presented as mean ± standard deviation (SD), median (interquartile range), or number of patients (%). BMI, body mass index; NRS, numeric rating scale; ASM, appendicular skeletal muscle mass; SMI, skeletal muscle mass index; HGS, handgrip strength.

**Table 2 jcm-11-05471-t002:** Sex-specific comparison of sarcopenia-related data between patients who reported a painDETECT score ≥ 13 (possible neuropathic-like pain symptoms) and those with a painDETECT score < 13 in patients with chronic musculoskeletal pain.

		Males			Females	
	PainDETECT	PainDETECT		PainDETECT	PainDETECT	
	≥13	<13	*p*-value	≥13	<13	*p*-value
	(n = 165)	(n = 894)		(n = 274)	(n = 1266)	
ASM, kg	23.24 ± 2.83	22.61 ± 2.80	0.077	14.63 ± 2.13	14.68 ± 2.17	0.810
SMI, kg/m^2^	8.00 ± 0.61	7.89 ± 0.66	0.166	5.98 ± 0.67	6.01 ± 0.67	0.693
HGS, kg	31.98 ± 10.73	33.38 ± 9.48	0.089	17.96 ± 5.99	18.78 ± 5.82	0.035

Values are presented as mean ± standard deviation (SD). ASM, appendicular skeletal muscle mass; SMI, skeletal muscle mass index; HGS, handgrip strength.

**Table 3 jcm-11-05471-t003:** Factors associated with neuropathic-like pain symptoms (painDETECT score ≥  13) in patients with chronic musculoskeletal pain: results from multivariate logistic regression analysis.

Variables	Adjusted OR	95% CI	*p*-Value
Female	1.251	0.814–1.924	0.308
Cardiovascular disease, yes	0.824	0.372–1.828	0.635
Mental health problems, yes	1.216	0.873–1.693	0.248
Pain duration, ≥12 months	0.536	0.390–0.738	<0.001
Pain score, NRS ≥ 7	1.329	0.946–1.866	0.101
Opioid usage, yes	2.961	1.860–4.714	<0.001
Main pain location			
Neck (cervical spine)	1.000		
Upper limbs (shoulder, elbow, wrist, hand)	1.464	0.780–2.750	0.236
Back (thoracic spine) and chest wall	1.447	0.757–2.766	0.264
Low back (lumbar spine)	0.904	0.564–1.448	0.675
Lower limbs (hip, knee, ankle, foot)	2.066	1.102–3.872	0.024
Sleep disturbance, yes	2.052	1.487–2.831	<0.001
ASM, kg	1.046	0.977–1.120	0.195
SMI, kg/m^2^	0.914	0.598–1.395	0.676
HGS, kg	0.976	0.960–0.993	0.005

OR, odds ratio; CI, confidence interval; NRS, numeric rating scale; ASM, appendicular skeletal muscle mass; SMI, skeletal muscle mass index; HGS, handgrip strength.

**Table 4 jcm-11-05471-t004:** Comparison of the results of the painDETECT items between patients with low and normal HGS #.

PainDETECT Items	Low HGS Group (n = 978)	Normal HGS Group (n = 1621)	*p*-Value
Gradation of pain (score of ≥3/5), n			
Burning sensation	113 (11.6)	180 (11.1)	0.725
Tingling sensation	424 (43.4)	706 (43.6)	0.921
Pain by light touch	85 (8.7)	105 (6.5)	0.036
Electric shock-like pain	326 (33.3)	519 (32.0)	0.488
Pain on cold/heat stimulation	54 (5.5)	45 (2.8)	<0.001
Numbness	178 (18.2)	249 (15.4)	0.058
Pain by slight pressure	222 (22.7)	345 (21.3)	0.397
Pain course pattern, n			
Persistent pain with slight fluctuation	491 (50.2)	884 (54.5)	0.032
Persistent pain with pain attacks	266 (27.2)	392 (24.2)	0.087
Pain attacks without pain between them	151 (15.4)	258 (15.9)	0.747
Pain attacks with pain between them	70 (7.2)	87 (5.4)	0.063
Radiating pain, n	387 (39.6)	572 (35.3)	0.029
Categorization by total score, n			
≤12	784 (80.2)	1376 (84.9)	
13–18	140 (14.3)	179 (11.0)	
≥19	54 (5.5)	66 (4.1)	
Total PainDETECT score	7.32 ± 6.28	6.73 ± 5.65	0.016

Values are presented as mean ± standard deviation (SD) or number of patients (%). HGS, handgrip strength. # The study population was divided into two groups: normal HGS (≥28 kg for men and ≥18 kg for women) and low HGS (<28 kg for men and <18 kg for women), according to the Asian Working Group for Sarcopenia (AWGS) 2019 guideline [1].

## Data Availability

Data are available upon request to corresponding author.

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
