# Peer review of "Neuropathic-like Pain Symptoms and Their Association with Muscle Strength in Patients with Chronic Musculoskeletal Pain"

_jcm, 2022, doi:10.3390/jcm11185471_

Round 1

Reviewer 1 Report

Comments to Authors:
This is an interesting article on a relevant topic. However, there are some aspects that need correction or review before publication:

Major issues:

1.      The Introduction section (Page 2/11, para 1): would benefit from adding a further description regarding the clinical significance of examining the impact of sarcopenia-related factors on the presence of neuropathic pain components in patients with chronic musculoskeletal pain. The lack of relevant research is not sufficient. In addition, it is important to present a few words in the Introduction section regarding the potential mechanistic association between neuropathic-like pain symptoms and their association with muscle strength.

2.      Different pain locations of chronic musculoskeletal muscle pain play a significant role in physical performance and adverse outcomes. Thus, it is critically important for you to provide further analyses regarding the impacts of different chronic musculoskeletal pain locations on neuropathic-like pain and their association with muscle strength.

3.      Line 110-111: Despite the appendicular skeletal muscle mass (ASM) being estimated using a previously anthropometric equation, we still doubt the actual validity of this equation. Reference 19 (Anthropometric equation for estimating appendicular skeletal muscle mass in Chinese adults. PMID: 22094840) developed the anthropometric equation based on 763 community-dwelling adults aged 18-69. Therefore, this prediction model may not apply to patients with chronic musculoskeletal pain. I recommend authors to address this important issue

4.      Line 122- Pain intensity was assessed using the average pain score on the numeric rating scale (NRS, 0 to 10) in the last 4 weeks. Examining the impacts of different pain intensities on primary outcomes was critical for risk stratification in clinical practice. Please add an explanation of why the threshold of 7 points was used for NRS categories in Table 1.

5.      Line 267-270- Theoretically, longer pain duration was associated with many adverse outcomes, especially for patients reporting chronic musculoskeletal muscle mass. While the finding was inconsistent with previous studies. Please provide a further discussion on this issue.

Author Response

We appreciate the kind and thorough review of our manuscript by the reviewers. We have made corrections in our manuscript in accordance with the reviewers’ comments. Attached herewith is our point-by-point responses to the reviewers’ comments.

We hope that our revised manuscript is now suitable for publication in Journal of Clinical Medicine. We look forward to hearing from you.

Reviewer 2 Report

I congratulate with the authors for their study, very interesting and original, and for the quality of presentation. The manuscript is indeed very clear and does not require many changes or editing. 

I just have a few questions: 

- The setting of the study is unclear. I guess patients were enrolled in a Pain Clinic, but more information on this issue should be included in the manuscript. Readers should be aware of the characteristics of subjects and expertise of clinicians that conducted the study. 

- Line 96: more detail is needed on how the cut-off point of painDETECT (13) was chosen and on whether it is applicable to the study population. 

- The main limitation of the study is represented by the absence of an instrumental evaluation of muscle mass. Although ASM and SMI were calculated with validated equations, no DXA was performed. On these bases, the diagnosis of sarcopenia is not certain, but only presumptive. I think this limitation should be better acnowledged and discussed in the manuscript. 

- No data on the causes of pain are reported in the manuscript. To better describe the characteristics of the study population, I suggest to introduce some data on the most frequent causes of pain in both patients with high and low painDETECT scores. 

Author Response

(The authors gave the same response as above.)

Reviewer 3 Report

In the present retrospective study, the authors investigated the association between sarcopenia, as identified by hand grip strength, and neuropathic-type pain as assessed by painDETECT questionnaire in a total of 2599 adult patients with musculoskeletal pain undergoing treatment for their pain. Overall, the work is interesting and of absolute relevance to research in this field as there is little evidence regarding the correlation between pain and sarcopenia. However, some important revisions should be addressed before publication. 

Abstract - This section is of paramount importance because it should entice the reader to read the paper in full. In this case, although the authors have described the contents of their manuscript in detail, it is in my opinion incomplete. In fact, it would be appropriate to add a background in a concise manner that introduces well the topic addressed in the manuscript. In addition, the abstract lacks a concluding sentence suggesting possible future perspectives that might emerge from the results obtained.

Introduction - This section should be significantly thorough and provide a broader perspective regarding sarcopenia and chronic musculoskeletal pain. It is important that it is immediately clear to the reader the importance of this study and what original insight it offers to the existing scientific literature. In general, the sentences are too concise and lack links.

 Lines 29-34: the authors briefly mention sarcopenia without mentioning its prevalence and associated social and health impacts.

 Lines 35-45: these focus on chronic pain, with particular attention to neuropathic pain. However, the authors do not provide any relationship between sarcopenia, which they have previously discussed, and chronic pain.

 Lines 38-40: The authors provide no definition of neuropathic pain and its impact on the population.

 Lines 43-45: They do not provide any useful information about the painDETECT questionnaire. In fact, the authors specify that this questionnaire is a valid tool for assessing the presence of neuropathic pain. What is the meaning and importance of the information in this sentence? The authors should describe and justify in detail the use of this questionnaire by also reporting recent data found in the literature.

 Lines 46-54: Chronic musculoskeletal pain is an extremely significant social problem, so the authors could provide information regarding the prevalence and socioeconomic burden of this condition. In this regard, the authors could refer to the recent publication by Bonanni et al (Bonanni R, Cariati I, Tancredi V, Iundusi R, Gasbarra E, Tarantino U. Chronic Pain in Musculoskeletal Diseases: Do You Know Your Enemy? J Clin Med. 2022 May 6;11(9):2609. doi: 10.3390/jcm11092609. PMID: 35566735; PMCID: PMC9101840) that summarizes in detail the topic at hand. Therefore, I suggest that the authors rework the entire section.

 Materials and Methods - Study population - Although in the next results section the authors provide demographic data of the study population, I suggest that the number of patients included in the study and their ages be included in this section as well.

 Results - The authors included patients between the ages of 20 and 98 in their study. This difference in age could significantly influence the results of the study as it is unlikely that the values of the parameters considered would be homogeneous between subjects as young as 20 and very old subjects over 90. The authors should consider dividing the patients included in the study by age groups as well and provide a summary table with related statistical analysis.

 Discussion - Undoubtedly, the writing of the discussion is more complete than that of the introduction as the authors have made an effort to provide a comparison with the existing literature. However, the information reported is sparse is lacking objective data that would allow a quantitative comparison between the authors' work and those cited. Therefore, the authors are strongly recommended to argue the reported evidence.

 Conclusions - It is important for this section to explicitly elucidate the added value of this study and what innovation it provides to the current literature.

Author Response

(The authors gave the same response as above.)

Round 2

Reviewer 1 Report

The author has answered all my issues. I don't have any more questions.

Author Response

We appreciate the kind and thorough review of our manuscript.

Reviewer 3 Report

The authors endeavoured to meet the Reviewer's requests.

Particularly, they fulfilled the request to provide a table offering a comparison between the different age groups, showing the absence of statistical significance between the groups. However, I feel that this information is important for the reader and suggest that the authors at least include this result as supplementary material.

Author Response

According to the reviewer’s suggestion, we were divided into the young group (20 to 44 years old), the middle-aged group (45 to 64 years old), and the old group (over 65 years old) and presented the results in Table 1 of the revised manuscript. 
